# OpenReview forum: "Revisiting Knowledge Tracing: A Simple and Powerful Model"
_ICLR.cc/2024/Conference — Submitted to ICLR 2024_

### Official Review · Reviewer_RjCq · 2023-10-29

**Soundness:** 3 good
**Presentation:** 3 good
**Contribution:** 3 good
**Rating:** 6
**Confidence:** 3

**Summary:**

This paper aims to address the knowledge state representation and the core architecture design challenges of knowledge tracing (KT). To this end, the authors propose the ReKT model. They first take inspiration from the decision-making process of human teachers and propose the knowledge state of students from three different perspectives. Then, the authors propose a Forget-Response-Update (FRU) framework as the core architecture for the KT task. They finally demonstrate the effectiveness of their model in terms of efficiency in computing and effectiveness in score prediction through experiments on 7 public real datasets. Their experimental results show that their proposed method can reach the best performance in the question-based KT task and the best/near-best performance in the concept-based KT task, and their proposal only requires 38% computing resources compared to other KT core architectures.

**Strengths:**

1. The paper introduces a multi-perspective approach to modeling the knowledge state of students, considering questions, concepts, and domains, which is logically self-consistent.

2. The FRU framework designed as the core architecture of ReKT is lightweight yet effective. According to experiments, ReKT can achieve competitive performance with significantly fewer parameters and computing resources compared to other core architectures.

3. The experimental results demonstrate the superior performance of ReKT in question-based KT tasks and its competitive performance in concept-based KT tasks, showcasing the effectiveness of the proposed model.

**Weaknesses:**

1. In terms of the methodology, the authors did not provide theoretical analysis about the spatial-temporal complexity of FRU. I hope the authors append such analysis to make the efficiency of their proposal in terms of computing resource more persuasive.

2. In terms of experiment, the authors only presented results in score prediction and computational resource cost. However, as the goal of the KT task is not only to predict students’ score sequences, but also to track the dynamic change of their knowledge states. Therefore, it will be helpful if the authors append such experiments (e.g., case study and visualization of student knowledge states) and use them to explain how their proposal can model student knowledge states better

**Questions:**

1. Can you further explain the design and workflow of the FRU framework? For example, what is the connection between FRU and human cognitive development models, what are inputs, what are learnable parameters and what are outputs? Besides, what does $I_\alpha$ mean in Section 3.4? I cannot find it on Figure 3.

2. In terms of sequence modeling, RNN and LSTM are also simple but effective. What are the advantages of FRU compared to them, especially in the context of knowledge tracing? Does your proposed FRU have the potential to be applied to other areas except for knowledge tracing (e.g., sequential recommendation)?

3. There are some syntax and spelling errors need to be solved, such as the index “I” in the formula of loss function, $Loss_{KT}$. I guess it should be replaced with $t$.

---

> ### Author Response · Authors · 2023-11-16
> **Response to Reviewer RjCq**
>
> Thank you for your detailed evaluation of our manuscript and for raising several insightful points. We appreciate the opportunity to address your concerns.
>
> **Q1**: “In terms of the methodology, the authors did not provide theoretical analysis about the spatial-temporal complexity of FRU. I hope the authors append such analysis to make the efficiency of their proposal in terms of computing resource more persuasive.”
>
> **A1**: Thank you very much for your suggestions! In Appendix G of this paper, we have added an analysis of the time and space complexity of FRU, along with the number of its learnable parameters.
>
> Once again, we sincerely appreciate your valuable advice!
>
> **Q2**: “In terms of experiment, the authors only presented results in score prediction and computational resource cost. However, as the goal of the KT task is not only to predict students’ score sequences, but also to track the dynamic change of their knowledge states. Therefore, it will be helpful if the authors append such experiments (e.g., case study and visualization of student knowledge states) and use them to explain how their proposal can model student knowledge states better”
>
> **A2**: Heartfelt thanks for your suggestions! We have added a case study comparing ReKT and other model for tracing students’ knowledge state in Appendix I, and analyzed the advantages brought by ReKT.
>
> Once again, we sincerely appreciate your valuable advice!
>
> **Q3**: “Can you further explain the design and workflow of the FRU framework? For example, what is the connection between FRU and human cognitive development models, what are inputs, what are learnable parameters and what are outputs? Besides, what does $I_\alpha$ mean in Section 3.4? I cannot find it on Figure 3.”
>
> **A3**: Thank you for your more in-depth questions. We have provided a detailed response to the question about FRU in the "General Response about FRU." It includes the motivation behind designing FRU, inspiration for its design, the process of FRU, and the differences and advantages of FRU compared to MLP, RNN, LSTM, and GRU.
>
> Furthermore, for FRU, its input consists of the student's historical interaction sequence (corresponding to $X_t$ in Figure 3). The learning parameters of FRU are the parameters within the Update and Forget components in Figure 3 (parameters $W_2$ and $b_2$ in Update, and parameters $W_1$ and $b_1$ in Forget). Its output is $Response_t$.
>
> Thank you very much for your thorough review. $I_\alpha$ represents the embedded representation of time interval $\alpha$ (For example, the time interval from the last time $\alpha=2$, assuming there is a time embedding matrix $I \in \mathbb{R}^{T \times d}$, $T$ is the total time step, $d$ is the embedding dimension, then $I_\alpha=I_2$ (that is, the $\alpha$-th row of $I$)).
>
> We have revised this detail in figure 3.($\alpha -> I_\alpha$)
>
> **Q4**: “In terms of sequence modeling, RNN and LSTM are also simple but effective. What are the advantages of FRU compared to them, especially in the context of knowledge tracing? Does your proposed FRU have the potential to be applied to other areas except for knowledge tracing (e.g., sequential recommendation)?”
>
> **A4**: Thank you for your insightful inquiry. We have provided a detailed response to the question about FRU in the "General Response about FRU." It includes the motivation behind designing FRU, inspiration for its design, the process of FRU, and the differences and advantages of FRU compared to MLP, RNN, LSTM, and GRU.
>
> Currently, we have not yet explored the application of FRU in other domains. However, given that FRU is inspired by models of human cognitive development, we believe there is theoretical potential for its application in domains related to modeling implicit states in humans, such as interests, health status, and more. The field you mentioned, sequential recommendation, is one that we consider to be a promising direction for potential application. Once again, we appreciate your insightful questions.
>
> **Q5**: “There are some syntax and spelling errors need to be solved, such as the index “I” in the formula of loss function, $Loss_{KT}$. I guess it should be replaced with $t$”
>
> **A5**: Thank you very much for your careful review. We have corrected this error in the paper.
>
> Thanks again for reviewing and your valuable suggestions!

---

> ### Author Response · Authors · 2023-11-21
> **Official Comment by Authors**
>
> Dear reviewer RjCq,
>
> As the window for reviewer-author interaction is closing soon, I wanted to extend my sincerest gratitude for the invaluable time and effort you have dedicated to reviewing our work. To ensure that we have met your expectations, may I kindly ask if you find our responses satisfactory and if there are any remaining issues that need further clarification or improvement?

---

> ### Author Response · Authors · 2023-11-23
> **Appreciation and Inquiry**
>
> We appreciate the valuable contributions you have made to our manuscript. As the rebuttal phase is about to close, we would like to inquire once again if we have adequately addressed your concerns and if you would be willing to consider improving our score.
>
> Once again, we sincerely appreciate the valuable time you dedicated to evaluating our work.

---

### Official Review · Reviewer_UX1Z · 2023-11-01

**Soundness:** 3 good
**Presentation:** 4 excellent
**Contribution:** 2 fair
**Rating:** 5
**Confidence:** 4

**Summary:**

The paper introduces a novel approach to Knowledge Tracing (KT) using the Forget-Response-Update (FRU) framework. KT is essential in online education systems for assessing and predicting student performance based on their interactions with educational content.

The FRU framework, designed based on human cognitive development models, stands out due to its lightweight nature, consisting of just two linear regression units. The proposed model, named ReKT, was extensively compared with 22 state-of-the-art KT models across 7 public datasets. Results demonstrated that ReKT consistently outperformed other methods, especially in question-based KT tasks. In concept-based KT tasks, an adapted version of ReKT, termed ReKT-concept, achieved top or near-top performance across datasets.

Furthermore, despite its simplicity, the FRU framework required only about 38% of the computing resources of other architectures like Transformers or LSTMs, showcasing its efficiency. The paper underscores the effectiveness, scalability, and efficiency of the FRU design in the realm of Knowledge Tracing.

**Strengths:**

The introduction of the Forget-Response-Update (FRU) framework offers a fresh perspective in the realm of Knowledge Tracing. While many models in the literature focus on complex architectures, the FRU's simplicity, relying on just two linear regression units, stands out as a unique proposition. The research brings a blend of cognitive learning principles and machine learning, fostering a more holistic approach to Knowledge Tracing.

The empirical evaluation of the proposed ReKT model is thorough. By benchmarking against 22 state-of-the-art KT models across 7 public datasets, the authors ensure a comprehensive assessment of their model's performance. The paper's methodological rigor is evident in the detailed descriptions of the FRU framework, the equations used, and the training methodologies employed.

The paper is well-structured, with distinct sections dedicated to introducing the problem, presenting the methodology, showcasing results, and discussing implications. The inclusion of figures, tables, and illustrative examples enhances the reader's understanding and provides a visual representation of the model's performance and capabilities.

**Weaknesses:**

The core of the proposed Forget-Response-Update (FRU) framework seems to be composed of two linear regression units. If this can be easily mirrored or replicated using two multi-layer perceptrons (MLPs), then the novelty of the FRU framework can be challenged. A deeper exploration or comparison with simple neural architectures, like MLPs, would provide clarity on the unique advantages of the FRU.

The use of terminology like "Forget", "Response", and "Update" in naming the modules of the FRU framework may imply distinct, targeted functionalities. However, in complex learning scenarios, such naming conventions can be misleading. In intricate neural architectures, a module named "Forget" might not necessarily perform a straightforward forgetting operation but might instead learn a more nuanced or intermediate representation. Over-reliance on such naming can lead to misconceptions about the actual functions and complexities of the modules, especially for those looking to adapt or build upon the framework.

While the lightweight nature of the FRU is emphasized, there's limited exploration on how the FRU can be integrated into or combined with deeper or more complex neural network architectures.

**Questions:**

How does the Forget-Response-Update (FRU) framework differ fundamentally from a structure consisting of two multi-layer perceptrons (MLPs)? What advantages does the FRU bring over a simple MLP setup?

Given the naming conventions like "Forget", "Response", and "Update", can you provide deeper insights into the exact functionalities and representations learned by each module during complex learning schedules?

How does the FRU framework integrate into more complex neural network architectures? Have there been experiments or considerations in this direction?

The paper mentioned that the FRU requires only about 38% of the computing resources compared to architectures like Transformers. Could you delve deeper into the parameter distribution within the FRU? Which module (Forget, Response, Update) consumes the most parameters?

---

> ### Author Response · Authors · 2023-11-16
> **Response to Reviewer UX1Z(1/2)**
>
> Thank you for your detailed evaluation of our manuscript and for raising several insightful points. We appreciate the opportunity to address your concerns.
>
> **Q1**: “The core of the proposed Forget-Response-Update (FRU) framework seems to be composed of two linear regression units. If this can be easily mirrored or replicated using two multi-layer perceptrons (MLPs), then the novelty of the FRU framework can be challenged. A deeper exploration or comparison with simple neural architectures, like MLPs, would provide clarity on the unique advantages of the FRU. How does the Forget-Response-Update (FRU) framework differ fundamentally from a structure consisting of two multi-layer perceptrons (MLPs)? What advantages does the FRU bring over a simple MLP setup?”
>
> **A1**: Thank you very much for your suggestion.FRU is a sequence model that involves forgetting and updating the knowledge state at each time step, similar to a recurrent neural network (RNN). On the other hand, MLP is a model for processing structured data.
>
> FRU is characterized by two linear regression units, and replacing these units with MLP or Attention, among other options, is possible (similar to replacing the four gates of an LSTM with four MLPs). However, we want to explain that the key to FRU is not the two linear regression units (it just illustrates the lightweight characteristics of FRU), but the ideas reflected in F, R, and U of FRU. We have provided a detailed response to the question about FRU in the "General Response about FRU." It includes the motivation behind designing FRU, inspiration for its design, the process of FRU, and the differences and advantages of FRU compared to MLP, RNN, LSTM, and GRU.
>
> In addition, during the experiment, we tried to replace the two linear regression units with two MLPs, and the performance is shown in the table below.
>
> | | ASSIST09| ASSIST09|ASSIST12|ASSIST12|ASSIST17|ASSIST17|Statics2011|Statics2011|EdNet|EdNet|Eedi|Eedi|
> | :----: | :----: | :----: | :----: | :----: | :----: | :----: | :----: | :----: | :----: | :----: | :----: | :----: |
> | **Evaluate**| AUC| ACC |AUC| ACC |AUC| ACC |AUC| ACC |AUC| ACC |AUC| ACC |
> | ReKT(MLP) | 0.7912| 0.7438|0.7852|0.7612|0.7818|0.7108|0.8972|0.8563|0.7768|0.7437|0.7972|0.7396|
> | ReKT    | 0.7917| 0.7449	     |0.7852|0.7609|0.7814|0.7102|0.8967|0.8568|0.7752|0.7447|0.7971|0.7397|
>
> It can be observed that whether using linear regression units or MLPs, the performance is nearly equivalent. This indicates that simple linear regression units are sufficient. Considering the spirit of simplifying the model as much as possible in this paper, we chose to use linear regression units.
>
> Once again, we sincerely appreciate your advice.
>
> **Q2**: “The use of terminology like "Forget", "Response", and "Update" in naming the modules of the FRU framework may imply distinct, targeted functionalities. However, in complex learning scenarios, such naming conventions can be misleading. In intricate neural architectures, a module named "Forget" might not necessarily perform a straightforward forgetting operation but might instead learn a more nuanced or intermediate representation. Over-reliance on such naming can lead to misconceptions about the actual functions and complexities of the modules, especially for those looking to adapt or build upon the framework.”
>
> **A2**: Thank you for your reminder. When naming FRU, we drew inspiration from the nomenclature of neural networks such as LSTM, GRU, as well as common module names in knowledge tracing methods like LPKT[1] and LBKT[2], such as "forget gate" and "update gate." Therefore, we named FRU as Forget-Response-Update. The issue you pointed out is indeed meaningful and may be a common challenge that deep learning methods find difficult to address. In future research, we plan to delve into the mechanistic roles and interdependencies of Forget-Response-Update in FRU. This promises to be an intriguing and vital area of exploration.
>
> Once again, we appreciate your insights.
>
> [1] Shuanghong Shen, Qi Liu, Enhong Chen, Zhenya Huang, Wei Huang, Yu Yin, Yu Su, and Shijin Wang. 2021. Learning process-consistent knowledge tracing. In Proceedings of the 27th ACM SIGKDD International Conference on Knowledge Discovery & Data Mining. Association for Computing Machinery, 1452–1460.
>
> [2] Bihan Xu, Zhenya Huang, Jiayu Liu, Shuanghong Shen, Qi Liu, Enhong Chen, Jinze Wu, and Shijin Wang. 2023. Learning behavior-oriented knowledge tracing. In Proceedings of the 29th ACM SIGKDD International Conference on Knowledge Discovery & Data Mining. Association for Computing Machinery, 2789–2800.

---

> ### Author Response · Authors · 2023-11-16
> **Response to Reviewer UX1Z(2/2)**
>
> **Q3**: “Given the naming conventions like "Forget", "Response", and "Update", can you provide deeper insights into the exact functionalities and representations learned by each module during complex learning schedules?”
>
> **A3**: Thank you for your further inquiry. In our design, for the forgetting process, we employ a linear forgetting model:
>
> $Response_t=f_t*Z_{t-\alpha}$
>
> where we reduce the state of knowledge accordingly, with $f_t \in [0,1]$. On the other hand, the updating process is described by an incremental model：
>
> $Z_t=Response_t+Tanh([Response_t \oplus X_t ]W_2+b_2)$
>
> This is used to model the changes in $Response_t$ after being stimulated by $X_t$. In our model design, we aimed to reference and emulate models of human cognitive development as much as possible.
>
> However, in the actual operational process of the model, the exact functions and representations learned by each module of FRU in complex learning scenarios may require the assistance of model interpretability methods to obtain clearer descriptions. Currently, we have not undertaken relevant work in this area, but we believe that a thorough analysis of the mechanisms and interdependencies of Forget-Response-Update in FRU will be an interesting and important topic.
>
> **Q4**: “While the lightweight nature of the FRU is emphasized, there's limited exploration on how the FRU can be integrated into or combined with deeper or more complex neural network architectures. How does the FRU framework integrate into more complex neural network architectures? Have there been experiments or considerations in this direction?”
>
> **A4**: Thank you for your more in-depth question. Currently, we have not undertaken the research of integrating FRU into more complex models. However, integrating FRU into other methods or expanding FRU is easy to implement. Specifically, FRU takes a historical sequence as input and outputs the student's current knowledge state, making it easy to use as a component in stacked models.
>
> Furthermore, expanding FRU is also simple. As mentioned in your earlier question, it is entirely feasible to replace FRU's two linear regression units with two MLPs. At the same time, FRU considers the time factor of forgetting by embedding the interval time and then transforming it. To extend FRU, one can explore modifications such as adjusting the computation method for the time factor in forgetting to incorporate exponential time decay [3].
>
> **Q5**: “The paper mentioned that the FRU requires only about 38% of the computing resources compared to architectures like Transformers. Could you delve deeper into the parameter distribution within the FRU? Which module (Forget, Response, Update) consumes the most parameters?”
>
> **A5**: Thank you very much for your suggestions! In Appendix G of the paper, we have added an analysis of the time and space complexity of FRU, along with the number of its learnable parameters.
>
> For convenience, assuming a consistent hidden layer dimension of $d$ in FRU, the number of learnable parameters for the forgetting module is $2d^2+d$, for the updating module is $2d^2+d$, and the response module has no learnable parameters.
>
>
> Thanks again for reviewing and your valuable suggestions!
>
> [3] Aritra Ghosh, Neil Heffernan, and Andrew S Lan. 2020. Context-aware attentive knowledge tracing. In Proceedings of the 26th ACM SIGKDD International Conference on Knowledge Discovery & Data Mining. Association for Computing Machinery, 2330–2339.

---

> ### Author Response · Authors · 2023-11-21
> **Official Comment by Authors**
>
> Dear reviewer UX1Z,
>
> As the window for reviewer-author interaction is closing soon, I wanted to extend my sincerest gratitude for the invaluable time and effort you have dedicated to reviewing our work. To ensure that we have met your expectations, may I kindly ask if you find our responses satisfactory and if there are any remaining issues that need further clarification or improvement?

---

> ### Author Response · Authors · 2023-11-23
> **Appreciation and Inquiry**
>
> We appreciate the valuable contributions you have made to our manuscript. As the rebuttal phase is about to close, we would like to inquire once again if we have adequately addressed your concerns and if you would be willing to consider improving our score.
>
> Once again, we sincerely appreciate the valuable time you dedicated to evaluating our work.

---

### Official Review · Reviewer_5ixm · 2023-11-01

**Soundness:** 3 good
**Presentation:** 3 good
**Contribution:** 3 good
**Rating:** 6
**Confidence:** 5

**Summary:**

This paper presents an improvement of the deep knowledge tracing (DKT) algorithm, ReKT. The authors revisited the DKT algorithm to design it from three perspectives.: 1)question: whether the question was attempted before, 20 concept: performance on questions with similar concepts, and 3) the entire trajectory.
Empirical results demonstrate the superior performance of ReKT compared to other variations of DKT.

**Strengths:**

- Superior performance while 38% less resource usage

**Weaknesses:**

1. The paper employs similar approaches to previous DKT methods, such as RAKT [1], AKT [2], and [3]  except for the FRU unit. All of three papers also implemented the FRU unit with exponential time decay as part of the attention mechanism in the transformer architecture. The authors have used MLP units as FRU and concatenated the hidden state to the final representations.

2. The authors did not provide any interpretations of the model's performance---which is very important in educational settings for both students' and teachers' perspectives. From a student's perspective, interpretation can help in recommending learning materials. From a teacher's perspective, it can be helpful to identify which questions or concepts students are struggling with.


References.
1. Pandey, S. and Srivastava, J., 2020, October. RKT: relation-aware self-attention for knowledge tracing. In Proceedings of the 29th ACM International Conference on Information & Knowledge Management (pp. 1205-1214).
2. Ghosh A, Heffernan N, Lan AS. Context-aware attentive knowledge tracing. InProceedings of the 26th ACM SIGKDD international conference on knowledge discovery & data mining 2020 Aug 23 (pp. 2330-2339).
3. Farhana E, Rutherford T, Lynch CF. Predictive Student Modelling in an Online Reading Platform. InProceedings of the AAAI Conference on Artificial Intelligence 2022 Jun 28 (Vol. 36, No. 11, pp. 12735-12743)

**Questions:**

The Rasch difficulty is determined from students' question and response binary matrix.

As the authors have three different representations of question interactions, did the authors compute the Rasch difficulty from three different interaction matrices?

How did the authors handle multiple submissions for computing the Rash difficulty?

---

> ### Author Response · Authors · 2023-11-16
> **Response to Reviewer 5ixm**
>
> Thank you for your detailed evaluation of our manuscript and for raising several insightful points. We appreciate the opportunity to address your concerns.
>
> **Q1**: “The paper employs similar approaches to previous DKT methods, such as RAKT [1], AKT [2], and [3] except for the FRU unit. All of three papers also implemented the FRU unit with exponential time decay as part of the attention mechanism in the transformer architecture. The authors have used MLP units as FRU and concatenated the hidden state to the final representations.”
>
> **A1**: Because the goal of this paper is to make the model both simple and effective. Among the KT models based on deep learning, the DKT model as the founder can be said to be very simple. The similarity between ReKT (the model proposed in this paper) and DKT demonstrates the simplicity of ReKT. Additionally, ReKT's three different perspectives (question, concept, domain) in modeling ensure its performance advantage.
>
> As you mentioned, the exponential decay method used in considering forgetting in the three models is indeed very effective, but this computational forgetting method is more complex [1]. This paper wants to simplify the operation as much as possible, so FRU's forgetting is not designed to have an exponential time decay effect. Of course, we believe that considering exponential decay would definitely make FRU more effective. The fact that FRU consists of two linear regression units and connects the hidden state to the final representation is to make the model as simple as possible.
>
> We also have provided a detailed response to the question about FRU in the "General Response about FRU". It includes the motivation behind designing FRU, inspiration for its design, the process of FRU, and the differences and advantages of FRU compared to MLP, RNN, LSTM, and GRU.
>
> **Q2**: “The authors did not provide any interpretations of the model's performance”
>
> **A2**: Thank you very much for reminding! We have added a case study of ReKT tracing students’ knowledge state in Appendix I.
>
> **Q3**: “As the authors have three different representations of question interactions, did the authors compute the Rasch difficulty from three different interaction matrices? How did the authors handle multiple submissions for computing the Rash difficulty?”
>
> **A3**: Thank you for your in-depth question. Since your two questions are similar, I will answer them both together.
>
> We did not calculate Rasch difficulty from three different interaction matrices. In our design, Rasch difficulty is specific to each question (i.e. each question has its own difficulty). Therefore, we fixed the representation of the question interactions, but the range seen in different perspectives is different, this allows the question interactions have different representations.
>
> The Rasch difficulty you pointed out should be for all questions. Then the difficulty of Rasch is different from different perspectives. This is a very interesting idea, and we have not yet considered this situation.
>
> But I have some ideas for your reference. Because the sequences obtained from different perspectives are different, their corresponding Rasch difficulty is also different. Then we can calculate the Rasch difficulty of the corresponding sequence from the question、concept、domain perspective respectively.
>
> If the inputs from the three perspectives are made consistent, then the interaction representation method needs to be modified (for example, the interaction representation can be changed to question + concept + question sequence Rasch difficulty + concept sequence Rasch difficulty + domain sequence Rasch difficulty).
>
> If the inputs from the three perspectives are inconsistent, then the input from the question perspective can be (question + question sequence Rasch difficulty), the input from the concept perspective can be (concept + concept sequence Rasch difficulty), and the input from the domain perspective can be (question + concept + domain sequence Rasch Difficulty).
>
> We hope our response has addressed your concerns. If you have any further questions, please feel free to let us know.
>
> Thanks again for reviewing and your valuable suggestions!
>
> [1] Zitao Liu, Qiongqiong Liu, Jiahao Chen, Shuyan Huang, and Weiqi Luo. simplekt: a simple but tough-to-beat baseline for knowledge tracing. In International Conference on Learning Representations, 2023b.

---

> ### Author Response · Authors · 2023-11-21
> **Official Comment by Authors**
>
> Dear reviewer 5ixm,
>
> As the window for reviewer-author interaction is closing soon, I wanted to extend my sincerest gratitude for the invaluable time and effort you have dedicated to reviewing our work. To ensure that we have met your expectations, may I kindly ask if you find our responses satisfactory and if there are any remaining issues that need further clarification or improvement?

---

> > ### Comment · Reviewer_5ixm · 2023-11-22
> >
> > I would like to thank all authors for their detailed responses. I do not have further questions at this point.

---

> ### Author Response · Authors · 2023-11-23
> **Appreciation and Inquiry**
>
> We are encouraged by the resolution of your concerns and would like to respectfully inquire if you would be willing to consider an increase in our score.
>
> Once again, we sincerely appreciate the valuable time you dedicated to evaluating our work.

---

### Official Review · Reviewer_c4BU · 2023-11-03

**Soundness:** 2 fair
**Presentation:** 3 good
**Contribution:** 2 fair
**Rating:** 5
**Confidence:** 4

**Summary:**

The paper proposes a simple yet power knowledge tracing (KT) model called ReKT. The method consists of 1) three levels of knowledge state modeling including question-, concept-, and domain-level, and 2) a forget-response update (FRU) unit. Extensive experiments show that ReKT achieves state of the art KT performance on an array of datasets comparing many baselines.

**Strengths:**

- The proposed method is claimed to be simple yet powerful.
- the evaluation appears to be comprehensive.

**Weaknesses:**

1. I am not convinced that the FRU gate is "simple". It appears to me as a variant of the gated recurrent unit (GRU) without the reset gate. Compared to the GRU, FRU has a similar forget gate and the hyperbolic tangent function in the end (without the affine combination of the update gate). I think the FRU architecture design, though interesting, does not qualify it as "very lightweight, as it consists of only two linear regression units". Otherwise, I can make the same "very lightweight" statement for GRU, which only consists of three linear regression units. Why not just use GRU? GRU takes into account not just forgetting (as in FRU), but also remembering/resetting, which might make more sense and have more modeling power? What exactly in the reference article "Toward a theory of instruction" do the authors get the inspiration to build FRU? This is an important question that the author should answer because they claim FRU as one of their core contributions, whereas I think FRU is not much different from GRU, which diminishes the value of this contribution. The authors should also cite GRU as important alternative modeling choices to compare to FRU (in addition to LSTM).

2. The proposed approach to represent knowledge at question, concept, and history level is not entirely new; methods such as learning factor analysis (https://link.springer.com/chapter/10.1007/11774303_17), performance factor analysis (https://files.eric.ed.gov/fulltext/ED506305.pdf), additive factor models (http://www.cs.cmu.edu/~ggordon/chi-etal-ifa.pdf, ), knowledge factoring machines (https://arxiv.org/pdf/1811.03388.pdf) also take into account of modeling students' knowledge at concept (sometimes called skills in these literature), question, or entire history levels. In the spirit of "revisiting", the authors neither mentioned nor compared to these classic knowledge tracing methods.

3. Some of the results need more clarifications. For example, AKT-R (https://arxiv.org/pdf/2007.12324.pdf) can achieve an AUC of __0.8346__ on ASSIST09 (see Table 5 in the AKT paper), beating the AUC of 0.7917 by the proposed method. Several other baselines in the AKT paper also achieve AUC > 0.8.

**Questions:**

1. How is FRU different from GRU? What motivate the differences?
2. How is the proposed method contexualized within, and compared to, some classic literature such as LFA, PFA, AFM, KFM, and others?
3. Why are some results different (sometimes by a large margin) to existing published results?

---

> ### Author Response · Authors · 2023-11-16
> **Response to Reviewer c4BU(1/2)**
>
> Thank you for your detailed evaluation of our manuscript and for raising several insightful points. We appreciate the opportunity to address your concerns.
>
> **Q1**: “I am not convinced that the FRU gate is "simple". It appears to me as a variant of the gated recurrent unit (GRU) without the reset gate. Compared to the GRU, FRU has a similar forget gate and the hyperbolic tangent function in the end (without the affine combination of the update gate). I think the FRU architecture design, though interesting, does not qualify it as "very lightweight, as it consists of only two linear regression units". Otherwise, I can make the same "very lightweight" statement for GRU, which only consists of three linear regression units.”
>
> **A1**: FRU is lightweight compared to most methods in the KT field (i.e. the lightweight nature of FRU is relative). FRU's design is also intended as a core framework for KT tasks. As mentioned in this paper, the current mainstream in KT research is building more complex models (such as stacked Transformers) to improve performance. In terms of being the core architecture for KT tasks, both FRU, GRU, and LSTM can be considered lightweight.
>
> **Q2**：” Why not just use GRU? GRU takes into account not just forgetting (as in FRU), but also remembering/resetting, which might make more sense and have more modeling power? What exactly in the reference article "Toward a theory of instruction" do the authors get the inspiration to build FRU? How is FRU different from GRU? What motivate the differences?”
>
> **A2**: Thank you for your in-depth questions about FRU. The purpose of designing FRU is to create a simpler (compared to LSTM, GRU, Transformer) architecture and suitable for KT, so we have not considered using GRU.
>
> Through experimentation, we found that we can achieve comparable or even better performance without the need for more complicated operations like memory/reset. Therefore, we simplified such operations.
>
> We have provided a detailed response to the question about FRU in the "General Response about FRU". It includes the motivation behind designing FRU, inspiration for its design, the process of FRU, and the differences and advantages of FRU compared to MLP, RNN, LSTM, and GRU.
>
> **Q3**: “The authors should also cite GRU as important alternative modeling choices to compare to FRU (in addition to LSTM).”
>
> **A3**: Thank you very much for your suggestion. We have added a comparison of GRU as the core architecture of KT in the paper (refer to Table 6, 7, 8, 9).
>
> **Q4**: “The proposed approach to represent knowledge at question, concept, and history level is not entirely new; methods such as LFA, PFA, AFM, KTM also take into account of modeling students' knowledge at concept (sometimes called skills in these literature), question, or entire history levels.”
>
> **A4**: Firstly, it needs to be clarified that classic knowledge tracing models based on machine learning, such as BKT, LFA, PFA, AFM, etc., only model student knowledge at the concept level. Knowledge tracing models based on machine learning, including KTM, only model student knowledge at the question level. They all struggle to utilize the entire interaction history of students (i.e., domain level). In contrast, mainstream approaches of deep learning-based knowledge tracing models utilize the entire interaction history of students (i.e., only domain level) to construct student knowledge (as more data helps improve the performance of deep learning models). However, very few methods exist that use deep learning to construct student knowledge from concept or question levels, which is evident as these perspectives imply a reduction in data volume (which is not beneficial to the performance of deep learning models). To the best of our knowledge, there is currently no knowledge tracing method that combines the three different perspectives of question, concept, and domain to construct student knowledge. Therefore, the proposed method in this paper is novel.
>
> **Q5**: “In the spirit of "revisiting", the authors neither mentioned nor compared to these classic knowledge tracing methods. How is the proposed method contexualized within, and compared to, some classic literature such as LFA, PFA, AFM, KFM, and others?”
>
> **A5**: Thank you very much for your reminder. We have added an review to the classic knowledge tracing method (BKT、LFA、PFA、AFM、KTM) in RELATED WORK. In addition, we added performance comparisons with classical knowledge tracing methods BKT and KTM (refer to Table 2, 3).

---

> ### Author Response · Authors · 2023-11-16
> **Response to Reviewer c4BU(2/2)**
>
> **Q6**: “Why are some results different (sometimes by a large margin) to existing published results”
>
> **A6**: This is mainly due to the way the data is processed. Suppose there is a student answer record $(q, (c_1, c_2, c_3))$: corresponding to a question $q$ and three concepts $(c_1, c_2, c_3)$ associated with it. According to [1], the processing method of this paper is to merge these three concepts $(c_1, c_2, c_3)$ into a single concept $c$, then in question-based KT, this record becomes $(q, c)$. What AKT does is to separate these three concepts $(c_1, c_2, c_3)$ into $(c_1)$, $(c_2)$, $(c_3)$. Then in question-based KT, this record becomes $(q, c_1), (q, c_2), (q, c_3)$ three records. In a real learning environment, the concept related to a specific question should remain unchanged. The processed approach in AKT cannot guarantee this, but the processed approach used in this paper can.
>
> Furthermore, the AKT performance (and other baselines) reported in recent papers at top-level conferences (such as simpleKT[2], LBKT[3], SFKT[4]) all indicate that the performance of AKT is not as high as reported in their paper.
>
> Thanks again for reviewing and your valuable suggestions!
>
> [1] Xiaolu Xiong, Siyuan Zhao, Eric G Van Inwegen, and Joseph E Beck. Going deeper with deep knowledge tracing. In Proceedings of the 9th International Conference on Educational Data Mining, pp. 545–550. International Educational Data Mining Society, 2016.
>
> [2] Zitao Liu, Qiongqiong Liu, Jiahao Chen, Shuyan Huang, and Weiqi Luo. simplekt: a simple but tough-to-beat baseline for knowledge tracing. In International Conference on Learning Representations, 2023b.
>
> [3] Bihan Xu, Zhenya Huang, Jiayu Liu, Shuanghong Shen, Qi Liu, Enhong Chen, Jinze Wu, and Shijin Wang. Learning behavior-oriented knowledge tracing. In Proceedings of the 29th ACM SIGKDD International Conference on Knowledge Discovery & Data Mining, pp. 2789–2800. Association for Computing Machinery, 2023.
>
> [4] Zhang M, Zhu X, Zhang C, et al. No Length Left Behind: Enhancing Knowledge Tracing for Modeling Sequences of Excessive or Insufficient Lengths. In Proceedings of the 32nd ACM International Conference on Information and Knowledge Management. 2023.

---

> ### Author Response · Authors · 2023-11-21
> **Official Comment by Authors**
>
> Dear reviewer c4BU,
>
> As the window for reviewer-author interaction is closing soon, I wanted to extend my sincerest gratitude for the invaluable time and effort you have dedicated to reviewing our work. To ensure that we have met your expectations, may I kindly ask if you find our responses satisfactory and if there are any remaining issues that need further clarification or improvement?

---

> ### Author Response · Authors · 2023-11-23
> **Appreciation and Inquiry**
>
> We appreciate the valuable contributions you have made to our manuscript. As the rebuttal phase is about to close, we would like to inquire once again if we have adequately addressed your concerns and if you would be willing to consider improving our score.
>
> Once again, we sincerely appreciate the valuable time you dedicated to evaluating our work.

---

### Author Response · Authors · 2023-11-16
**General Response about FRU**

**Motivation for designing FRU:**

From the DKT (LSTM architecture) onwards, researchers in the field of KT have continuously introduced models with stronger learning capabilities (such as Transformer) to improve the accuracy of KT tasks. This not only makes the latest KT models increasingly complex and computationally expensive, but also overlooks the exploration of the characteristics of the KT task itself. Therefore, this paper aims to revisit the core of the KT task itself, abandoning complex architectural designs, and designing a simple and applicable core architecture for KT. Compared to the current mainstream Transformer architecture (with multiple heads and layers), the early models based on LSTM (DKT) can be considered lightweight models. However, we hope to design an even more extremely lightweight model that combines the characteristics of KT.

**Inspiration for designing FRU:**

The key to the KT problem lies in modeling the knowledge state of students. Therefore, FRU draws on the human cognitive development model, which emphasizes that changes in human knowledge state (Response) are mainly influenced by two psychological processes: internalization and forgetting. Internalization describes the adjustment of the internal knowledge state based on environmental stimuli, emphasizing the updating (Update) process of knowledge state based on stimuli. Forgetting (Forget) describes the natural change of knowledge state over time, emphasizing the process of knowledge state changing over time.

**FRU:**

In FRU, we implement **forgetting** estimation of students’ knowledge state based on the interval time, **response** directly reacts based on student knowledge state, and finally **updates** the knowledge state based on the learning interaction.

**The differences and advantages between FRU and MLP, RNN, GRU, and LSTM**(We also explain this question in Appendix H of the paper)

**A.The differences between FRU and MLP:**

1.FRU is a sequence model, similar to recurrent neural networks such as RNN, while MLP is a model that processes structured data.

2.Both linear regression units of FRU have only one change, not multiple.

**B. The differences between FRU and RNN, GRU, and LSTM:**

1.FRU reduces the output transformation layer.

2.Compared with RNN, FRU considers the forgetting process at each moment; compared with LSTM and GRU, their gates (such as LSTM's forgetting and input gates) are connected in parallel; FRU's foget-response-update is in series.

3.When calculating the current state, LSTM additionally uses global state updates, and GRU makes additional changes. FRU, on the other hand, directly relies on the state of the previous moment without involving additional changes.

4.FRU considers the case of processing sequences with non-uniform time intervals.

5.FRU has fewer learnable parameters than LSTM and GRU.

**C. The Advantages of FRU:**

1.As the core architecture of KT, FRU is simpler than other methods and can maintain equivalent or better performance while using less computing resources.

2.FRU is able to handle sequences with non-uniform time intervals.

---

### Author Response · Authors · 2023-11-16
**General Response about the revised paper**

Thank you very much for all the suggestions provided by all the reviewers! We have made the following modifications in the paper, which are highlighted in blue font:
1. We have made some modifications to introducing FRU.
2. We have added a comparison with GRU as the core architecture for KT (refer to Table 6, 7, 8, 9).
3. In the RELATED WORK section, we have included a review of classic knowledge tracing models and added performance comparisons with classical knowledge tracing methods BKT and KTM (refer to Table 2, 3).
4. Appendix D now includes an analysis of the time and space complexity of FRU.
5. Appendix H has been updated with the differences and advantages of FRU over MLP, RNN, GRU, and LSTM.
6. Appendix I contains a case study of ReKT for tracing student knowledge states.
7. In Figure 3, we have changed $\alpha$ (time interval) to $I_\alpha$ (embedding representation of the time interval).
8. We have modified the spelling error of the subscript in the formula for calculating $Loss_{KT}$.

Additionally, due to space limitations in the main content of the revised paper (9 page), we have moved the original Table 4 (ablation experiments of ReKT-concept) to Appendix C.

Once again, we sincerely thank all the reviewers for their valuable help in improving the quality of our work!

---

### Author Response · Authors · 2023-11-23
**Thank you to all the reviewers**

The rebuttal phase is coming to a close, and we would like to express our gratitude to all the reviewers for their valuable comments and assistance with our paper.

Thank you.

---

### Meta-Review · Area_Chair_PJCJ · 2023-12-07

**Metareview:**

This paper proposes a simple and efficient sequential model structure that considers three levels of knowledge states in the knowledge tracing problem. The components, inspired by domain knowledge, have been shown to be experimentally effective, and the presentation and discussion have been greatly improved through revision. However, the technical contributions of the proposed ideas was not fully appreciated in the context of machine learning; eventually, the paper is around the borderline. Perhaps conferences in EDM/learning analytics area would have been more appreciative of the paper.

**Justification For Why Not Higher Score:**

While this paper provides useful insights into the knowledge tracing problem, its impact on the community is limited when considering the effectiveness of the proposed ideas in other areas.

**Justification For Why Not Lower Score:**

NA

---

### Decision · Program_Chairs · 2024-01-16

Reject